# Hadronic Interactions in CRPropa with state-of-the-art generators

**Leonel Morejon**[1]⋆

**1** Wuppertal University, Gaussstrasse 20, 42117 Wuppertal, Germany

⋆ leonel.morejon@uni-wuppertal.de

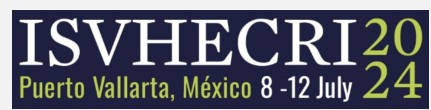

*22nd International Symposium on Very High Energy Cosmic Ray Interactions (ISVHECRI 2024) Puerto Vallarta, Mexico, 8-12 July 2024* doi:10.21468/SciPostPhysProc.?

## Abstract

CRPropa 3.2, released recently, is the latest update in a continued effort to maintain and extend this open-source code well known in the cosmic-ray community. Originally aimed at simulating the ballistic propagation and interactions of Ultra-High Energy Cosmic Rays (UHECRs), today it can handle diffusive propagation of cosmic rays in a variety of magnetic fields, model stochastic cosmic ray acceleration, simulate electromagnetic cascades for gamma ray emission and transport, and provides other capabilities. Of special interest is the recent introduction of a hadronic module to facilitate the treatment of cosmic ray interactions in the galaxy and within the sources. This work details the recent updates on this module in the context of bursting sources of UHECRs.

# 1  Introduction

Hadronic interactions of ultra-high-energy cosmic rays (UHECRs) are not only important for air showers, but they can also be relevant in the sources. During acceleration and transport in the sources, UHECRs interact primarily with photons from surrounding structures like the kilonova in gamma ray bursts, or thermal emission from accretion disks like in active galactive nuclei. Nevertheless, hadronic interactions are expected to play some role in some sources as has been shown in other works (e.g. [1,2]).

Previous treatments of hadronic interactions in astrophysical scenarios (e.g. [3]) are based on precomputed tables with limited types of secondaries produced. On one hand, it can be more efficient to perform Monte Carlo sampling from tables, and on the other, expected spectra can be numerically computed directly with such tables. Nevertheless, this method can have limitations as the range of applicability is determined by the range of energies, primary and secondary species, final state cutoff, generator version, etc. employed to construct the tables. This is evidenced in the Appendix A where the distribution of neutral pions obtained in reference [4] is compared to the distribution obtained with the more recent version QGSJet-II.04 [5]. Furthermore, the diversity of UHECR source scenarios and the need to access updated hadronic interaction generators (HIGs) for their evaluation in astrophysical contexts are best served with a direct access approach such as the HIM [6] developed for CRPropa [7].

Previously, the HIM for CRPropa was introduced in reference [6], detailing its structure, estimating its efficiency and showcasing proton-proton interactions in an example consistent with a bursting source of UHECRs. This contribution discusses the inclusion of proton-nucleus interactions, the treatment of secondary nuclear fragments and the impact on the simulation of a typical bursting source of UHECRs.

# 2  The Hadronic Interactions Module (HIM)

The HIM is a python module based on the available CRPropa class provided for the implementation of external modules, and it makes use of the frontend **chromo** [8] to generate hadronic interactions with the available generators (some of which are Epos-LHC [9], QGSJet-II.04 [10], Sibyll2.3c [11], and others). The module implements the tasks of deciding the success of a hadronic interaction, generating the secondary products and feeding the cinematic quantities of the products back to CRPropa for their propagation. Additional details on the HIM are given in [6] and the current release discussed here is [12].

## 2.1  Treatment of A-p interactions

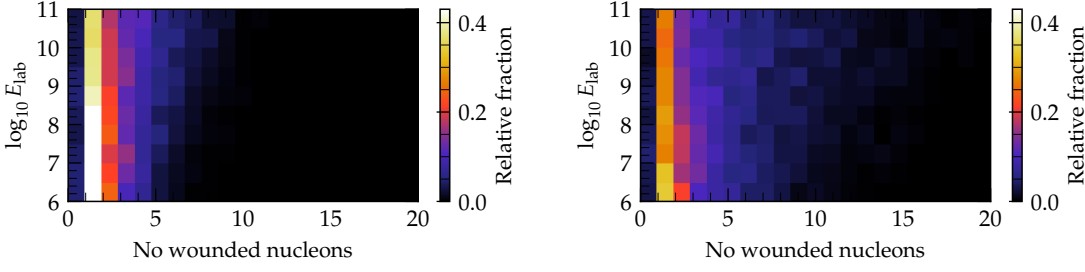

Figure 1: Wounded nucleon distributions with QGSJet-II.04 for different projectile nuclei on target protons: nitrogen (left) and iron (right).

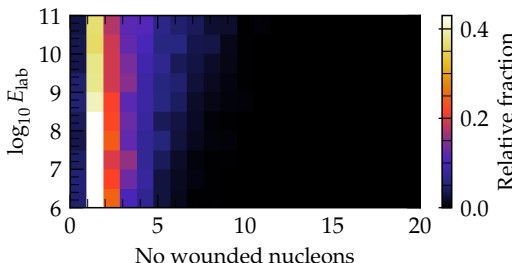 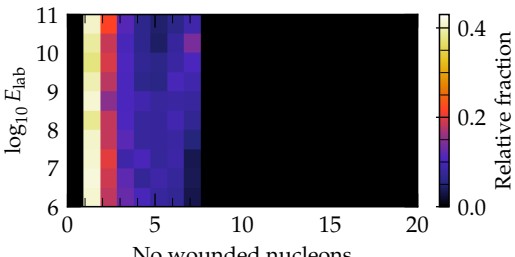

Figure 2: Wounded nucleon distributions for nitrogen projectile nuclei on target protons: QGSJet-II.04 (left) and EPOS-LHC (right).

43  The inelastic cross section in the HIM is available directly from the HIGs, however, more
44  efficient evaluations are provided to deal with simulations where a large number of interactions
45  are expected, or as alternative where the computation of the cross section by the HIGs takes
46  longer than desired. The default inelastic cross section $\sigma_{\mathrm{inel}}(E_{\mathrm{lab}})$ is estimated as a constant
47  fraction of the total cross section $\sigma_{\mathrm{tot}}$

$$\sigma_{\mathrm{inel}}(E_{\mathrm{lab}}) = 0.81 \cdot \sigma_{\mathrm{tot}}(E_{\mathrm{lab}}) \tag{1}$$

48  evaluated from the parametrization in reference [13]. Alternatively, the cross sections are
49  obtained from precomputed values with the respective HIG.
50  The nuclear fragments produced in an event are part of the output of most HIGs. How-
51  ever, at the moment of this work, **chromo** does not provide a utility function to access this
52  information consistently between hadronic models (see issue #183 on the github repository
53  of **chromo** [8]). As a fallback method, and for consistency between HIGs, the HIM computes
54  the nuclear remnant fragments based on the number of wounded nucleons, estimating the
55  mass of the remnant $A_r$ as the number of spectator nucleons $A_r = A - w_n$, with $A$ the mass
56  of the target nucleus and $N_w$ the number of wounded nucleons. The distribution of $N_w$ as a
57  function the projectile lab. energy (see Figure 1) has a very weak dependence on the energy,
58  with a modest broadening towards the largest energies. The effect of the projectile mass is
59  more pronounced as the distributions are noticeably broader for iron in contrast to nitrogen.
60  The comparison between HIGs is illustrated in Figure 2, where distributions obtained with
61  QGSJet-II.04 tend to be narrower compared to EPOS-LHC, which are limited to at most six
62  nucleons. For both cases, the broadening with projectile energy is appreciable. The specific
63  nuclear species of the remnant is determined by randomly choosing the proton number to
64  match one of the nuclear species regarded as stable in CRPropa. The distribution of remnants
65  resulting from this prescription are shown in Figure 3, given for two different projectile nuclei
66  on proton targets: nitrogen projectiles (left) and iron projectiles (right).

## 3  Example: UHECR bursting source scenario

68  The fragment cascades described are illustrated in a benchmark example of bursting source
69  of UHECRs with the same physical parameters as in reference [6], for ease of comparison.
70  The benchmark source is represented by a blob of radius $R = 1\,\mathrm{pc}$ inside which the pro-
71  ton density is homogeneous and whose value is computed to yield a desired optical depth
72  as $\tau = \sigma_{\mathrm{inel}}(E_{\mathrm{lab}})\rho R$. The magnetic field follows a Kolmogorov distribution with a root-mean-
73  square intensity of $1\,\mathrm{G}$ and a coherence length of $0.17\,R$. The injected cosmic rays follow
74  flat energy spectrum in logarithmic scale ($\frac{dN}{dE}(E) \propto E^{-1}$) in the energy range 1-100 EeV. No
75  other interactions were included besides hadronic interactions of the injected nuclei and the

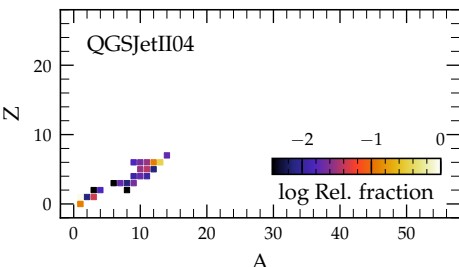 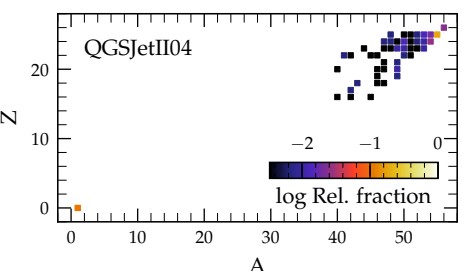

Figure 3: Fragment distributions for different projectile nuclei: nitrogen (left) and iron (right).

fragments produced, thus the decays of unstable fragments were not applied. As final state secondaries only photons, electrons, pions, neutrons and protons and their anti-particles were considered, however the HIM provides functionality to decide the list final state particles.

Figures 4 and 5 show the spectra of secondaries including the nuclear fragments resulting from the injection of helium and nitrogen primaries respectively. The most frequent nuclear remnants are shown grouped by nuclear mass as well as nucleons and antinucleons. In the case of 10% optical depth the injected nuclei experience very few interactions and the injected spectrum is barely changed unlike in the case of 100% optical depth. The spectra of light secondaries is mainly dominated by neutral pions while the other pions can be trapped by the magnetic field which is consistent with the previous results (see [6]). The spectra of pions and nucleons at lower energies contain contributions from nuclei of all energies, but at higher energies only the most energetic nuclei contribute as implied by the softer spectra in the 10% optical depth cases.

The current release for the HIM [12] is available for open usage by the community, in the framework of CRPropa. The detailed physical impact of these interactions are the subject of study in following publication currently in preparation.

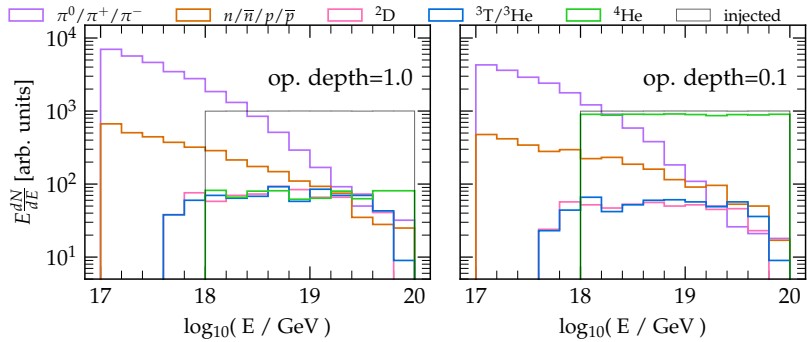

Figure 4: Spectra of escaping secondaries for two different proton densities with helium injection.

# Acknowledgements

This work has received funding from the DFG through the grant "MultI-messenger probe of Cosmic Ray Origins" (MICRO), project number 445990517 (KA 710/5-1).

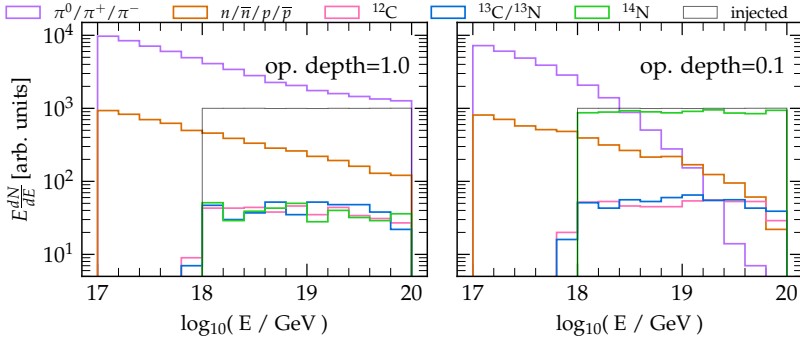

Figure 5: Spectra of escaping secondaries for two different proton densities with nitrogen injection.

## A    Limitations of precomputed tables

As illustration of one of the limitations of precomputed tables and fits to such tables, Figure 6 shows differences between a more recent version of QGSJet and a fit performed in [4] with an earlier version of the code. While the pion distributions for 100 GeV projectile protons seem in good agreement, the fitted distribution is biased toward larger pion energies and under predicts the low energies by about 30-40% for 1 PeV projectile protons.

The importance of these differences depend on the spectra of projectile protons under con- sideration. A recomputation of the tables and fits is always possible, however this added step is a disadvantage compared to the approach of direct sampling from generators employed in the HIM, which makes recent updated available upon release and avoids delays associated with the needed updates in the existing tables. Furthermore, direct sampling from the generators has a wider range of application because, unlike in precomputed tables, no assumptions are made on the energy range and specific secondaries of interest, whereas tables will are limited by these assumptions and can become prohibitively large if the range of desired applicability is too broad.

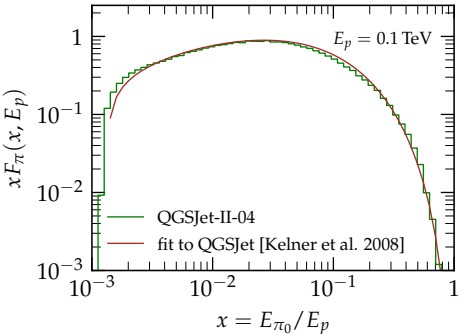
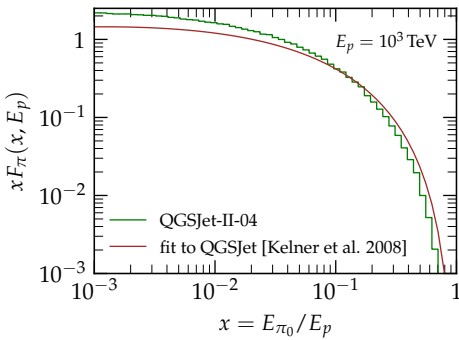

Figure 6: Comparison of the distributions of neutral pions as reported in reference [4] versus sampling in the HIM of the more recent QGSJet-II.04 [5].

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
