# Peer review of "Hadronic Interactions in CRPropa with state-of-the-art generators"

_SciPost Physics Proceedings_

## Round 1 · Referee Report · Anonymous (Referee 1) · 2025-1-28

Strengths

  • Concise and informative about a recent development for a major astroparticle simulation tool.

Weaknesses

  • Some details of the results (figures) could be discussed in more depth

Report

Report:
The contribution explains how hadronic (or rather nuclear?) interaction models have been implemented in the recent CRPropa update.

Remark: HIG hadronic interaction generators —> hadronic event generators

Question/concern about Fig. 5: How can the injection of nitrogen up to 100 EeV produce protons in the uppermost energy bin, or Helium nuclei? Helium secondaries can maximally reach < 30 EeV or a bit more with Fermi motion.

Figures 4 and 5: Is it a coincidence that all secondary elements have more or less the same distribution and weight or a coincidence? Highlight the response to these questions in the caption.

About Figure 6: The difference is somewhat expected since a QGSJet-II 04 result is compared with a fit to QGSJet-01. This is unrelated to the method, and the text should be slightly rephrased to reflect this fact. Or, if the comparisons of both methods are the main topic, then QGSJet-01 is also available in Chromo, and the figure could be redone with it if this is feasible.

Requested changes

See report

Recommendation

Ask for minor revision

---

## Editorial Decision

in_refereeing